# Using Statistical Parametric Mapping to Compare the Propulsion of Age-Group Swimmers in Front Crawl Acquired with the Aquanex System

**DOI:** 10.3390/s22218549

**Published:** 2022-11-06

**Authors:** Jorge E. Morais, Daniel A. Marinho, João P. Oliveira, Tatiana Sampaio, Tiago Lopes, Tiago M. Barbosa

**Affiliations:** 1Research Center in Sports, Health and Human Development (CIDESD), University of Beira Interior, 6201-001 Covilhã, Portugal; 2Department of Sport Sciences, Instituto Politécnico de Bragança, 5300-252 Bragança, Portugal; 3Department of Sport Sciences, University of Beira Interior, 6201-001 Covilhã, Portugal

**Keywords:** biomechanics, kinetics, performance, youth, propulsion, sensors

## Abstract

Understanding the difference in each upper limb between age groups can provide deeper insights into swimmers’ propulsion. This study aimed to: (1) compare swimming velocity and a set of kinematical variables between junior and juvenile swimmers and (2) compare the propulsion outputs through discrete and continuous analyses (Statistical Parametric Mapping—SPM) between junior and juvenile swimmers for each upper limb (i.e., dominant and non-dominant). The sample was composed of 22 male swimmers (12 juniors with 16.35 ± 0.74 years; 10 juveniles with 15.40 ± 0.32 years). A set of kinematic and propulsion variables was measured at maximum swimming velocity. Statistical Parametric Mapping was used as a continuous analysis approach to identify differences in the propulsion of both upper limbs between junior and juvenile swimmers. Junior swimmers were significantly faster than juveniles (*p* = 0.04, d = 0.86). Although juniors showed higher propulsion values, the SPM did not reveal significant differences (*p* < 0.05) for dominant and non-dominant upper limbs between the two age groups. This indicates that other factors (such as drag) may be responsible for the difference in swimming velocity. Coaches and swimmers should be aware that an increase in propulsion alone may not immediately lead to an increase in swimming velocity.

## 1. Introduction

Competitive swimming is a time-based sport where the main goal is to complete the given distance in the shortest possible time. The net balance between propulsive and drag forces acting on the swimmer determines the acceleration of the body and, therefore, the swimming velocity [1]. The literature reports significant experimental research on the drag of swimmers performing the front-crawl stroke [2,3,4]. Overall, such studies have observed that swimmers who have a better hydrodynamic position, namely by decreasing the frontal surface area, are more likely to present a faster swimming velocity [3,4]. On the other hand, there is less evidence on the swimmers’ propulsion measured directly, at least by experimental methods [5]. 

Nowadays, the use of wearables to monitor variables in swimming plays a key role [6]. This allows for more insightful information about phenomena that were more difficult to understand or measure in the past. In the case of swimming, the use of wearables allows researchers and practitioners to acquire propulsion data experimentally, which, in the past, were based on assumptions of numerical methods [7,8]. Tethered swimming was one experimental method used to measure swimmers’ propulsion [9,10]. However, it was argued that this may not be an accurate method to assess the swimmers’ propulsion. Being strapped at the hips may not accurately represent: (i) the effect of the propulsion on the swimmers’ center-of-mass displacement [11] and (ii) swimmers’ stroke pattern due to lack of displacement [12]. Therefore, measuring propulsion using wearable devices that allow swimmers to move “freely” while performing their own stroke pattern will provide better information about their propulsion [7]. In the front-crawl stroke, the propulsion generated by the upper limbs receives significant attention, as it is responsible for 90% of the total swimming velocity [13]. Overall, based on the use of such wearables, it was experimentally found that greater propulsion leads to faster swimming velocities [8,14]. Thus, it can be considered that faster swimmers generate higher levels of propulsion.

However, such assumptions were based on discrete variables, considering the average value. That is, researchers use the average value obtained during the full-stroke cycle [14,15]. The main issue with discrete data is that they diminish the dimensionality of potential information available by disregarding or not considering insightful details throughout the entire movement. Conversely, continuous analysis of time-series data through Statistical parametric Modelling (SPM) can be of paramount importance. This allows researchers to maintain the integrity and sensitivity of the time-series data, allowing them to understand what happens across the time-series [16]. In the case of swimming (particularly front crawl), this can be extremely helpful in understanding at what specific point in the arm pull faster swimmers generate higher values of propulsion than their slower counterparts. 

Therefore, this study aimed to: (i) compare swimming velocity and a set of kinematical variables between junior and juvenile swimmers and (ii) compare the propulsion outputs through discrete and continuous analyses (SPM) between junior and juvenile swimmers for each upper limb (i.e., dominant and non-dominant). It was hypothesized that junior swimmers would present faster swimming velocity than juvenile swimmers. Moreover, higher and more significant values of propulsion by both upper limbs would be verified in junior swimmers compared to their juvenile counterparts by discrete and continuous analyses. 

## 2. Materials and Methods

Figure 1 shows a flowchart of the experiments carried out from the swimmers’ recruitment to the data collection.

### 2.1. Participants

Participants were 22 male swimmers divided into two groups (junior level: N = 12, age = 16.35 ± 0.74 years, body mass = 70.64 ± 5.65 kg, height = 177.42 ± 5.14 cm, arm span = 182.96 ± 9.15 cm, frontal surface area = 0.1021 ± 0.01 m^2^, FINA points = 572.17 ± 67.32 at the 100 m freestyle event in short-course meter; juvenile level: N = 10, age = 15.40 ± 0.32 years, body mass = 66.88 ± 8.14 kg, height = 176.30 ± 6.46 cm, arm span = 182.50 ± 7.74 cm, frontal surface area = 0.1002 ± 0.01 m^2^, FINA points = 560.30 ± 43.72 at the 100 m freestyle event in short-course meter) regularly participating at regional- and national-level competition. The swimmers were recruited from a national swim squad. The inclusion criteria were: (i) swimmers had to be national-level swimmers in each age group in freestyle sprint events and (ii) participate regularly in the daily training sessions scheduled from the beginning of the season. Swimmers who had an injury that prevented them from performing daily training were excluded. They had more than five years of competitive swimming experience. The sample included age group national record holders and swimmers enrolled in a national talent-identification scheme (Tier 3) [17]. Competitive level groups were formed based on their competition age group, which is categorized based on the chronological age of the swimmers. Swimmers participated in six to nine training sessions per week (90 min for each session). Parents or guardians and swimmers provided informed consent. All procedures were in accordance with the Declaration of Helsinki regarding human research and the Polytechnic Ethics Board approved the research design (N. 72/2022).

### 2.2. Kinematic Data Collection

Prior to data collection, the swimmers performed a standardized warm-up orientated for sprint events [18]. Data were collected over 25 m all-out trials in front crawl with a push-off start. Swimmers were instructed to hold their breath between the 11th- and 24th-meter marks to avoid disruptions/changes in stroke coordination or technique. Each swimmer performed three trials with 10 min of recovery between each and the best one (i.e., the fastest) was used for further analysis. Three consecutive stroke cycles were analyzed between the 11th- and 24th-meter marks and the average was used for analysis. The string of a mechanical apparatus (SpeedRT, ApLab, Rome, Italy) was attached to the swimmer’s waist [19]. The speedometer calculated the displacement and velocity of the swimmer at a sampling rate of 100 Hz. Afterward, data were imported into signal-processing software (AcqKnowledge v. 3.9.0, Biopac Systems, Santa Barbara, CA, USA). The signal was handled with a Butterworth 4th-order low-pass filter (cut-off: 5 Hz) upon residual analysis. A video camera (FDR-X3000, Sony, Tokyo, Japan) synchronized with the mechanical apparatus recorded the swimmer’s displacement in the sagittal plane to identify the entry and exit of the hand in the water to indicate the stroke cycles. Swimming velocity (m·s^−1^) was retrieved from the software during three consecutive stroke cycles. The intra-cyclic fluctuation in swimming velocity (dv, in %) was calculated as the coefficient of variation (CV), whereby CV = one standard deviation/mean·100 [20]. Stroke frequency (SF, in Hz) was calculated as the number of cycles per unit of time (based on the footage recorded), from the time required to complete one full cycle (f = 1/P; where P is the period) and later converted to Hz. Stroke length (SL, in m) was calculated as SL = v/SF [21].

### 2.3. Propulsion Data Collection

Propulsion data were acquired simultaneously with kinematic data. Pressure sensors (Swimming Technology Research, Richmond, VA, USA) were used to measure propulsion at a sampling rate of 100 Hz [22]. On each hand, sensors were placed between the third and fourth metacarpals to measure the pressure differential between the palmar and dorsal surfaces. This location is considered a good proxy for the point of application of the propulsion vector for the hand [23]. Swimmers were asked to keep their hands immersed at a depth of 0.50 m for 10 s to calibrate the system. This procedure was performed at the beginning of each trial. The pressure sensor data were transferred to the Aquanex software (Aquanex v. 4.2 C1211, Richmond, VA, USA) by an A/D converter [24]. Time-propulsion series were imported into signal-processing software (AcqKnowledge v. 3.9.0, Biopac Systems, Santa Barbara, CA, USA) for signal handling using a Butterworth 4th-order low-pass filter (cut-off: 5 Hz). For the dominant (F_mean_dominant_, in N) and non-dominant (F_mean_non-dominant_, in N) arm pull, the mean propulsion was measured. The intra-cyclic variation in propulsion of each upper limb (dF_dominant_ and dF_non-dominant_, in %) was calculated as being the CV, as previously mentioned. Identification of the swimmers’ hand dominance was made by self-report as reported by others [24].

### 2.4. Statistical Analysis

The Shapiro–Wilk and Levene’s tests were used to assess the normality and homoscedasticity, respectively. Mean plus one standard deviation, mean differences between groups and 95% confidence intervals (95CI) were computed as descriptive statistics. Two-tailed independent samples *t*-tests (*p* < 0.05) were used to compare groups. Cohen’s d estimated the standardized effect sizes and was deemed as: (1) trivial if 0 ≤ d < 0.20; (2) small if 0.20 ≤ d < 0.60; (3) moderate if 0.60 ≤ d < 1.20; (4) large if 1.20 ≤ d < 2.00; (5) very large if 2.00 ≤ d < 4.00; (6) nearly perfect if d ≥ 4.00 [25].

SPM two-tailed independent sample *t*-tests (*p* < 0.05) were used to compare the propulsion–time series between groups (i.e., dominant upper limb: junior vs juvenile; non-dominant upper limb: junior vs juvenile) [16]. Prior to this analysis, each stroke cycle was normalized to its duration in an R routine [26]. Cycle normalization is a method used to obtain a uniform representation of any cycle (i.e., standardization) for the purpose of comparing or averaging data across participants. The usual method is based on representation of a percentage of the complete cycle. Otherwise, the cycles between subjects could not be directly compared, as they would present temporal differences. SPM analyses were implemented using the open-source spm1d code on Matlab (v.M0.1, www.spm1d.org (accessed on 15 July 2022)).

## 3. Results

Table 1 presents the descriptive data (mean ± one standard deviation) of the variables assessed for junior and juvenile swimmers. It also presents the *t*-test comparison between age groups (i.e., junior vs juvenile) for kinematical and propulsion variables. Swimming speed presented a significant difference with moderate effect size among age-group swimmers (mean difference = −0.063, t = −2.18, *p* = 0.04, d = 0.86), with juniors being the fastest swimmers. The same trend was observed for the SF. As for the propulsion variables, the F_mean_ of the dominant upper limb presented a significant difference with a moderate effect between age groups (i.e., junior vs juvenile). By contrast, non-significant differences (with small effect) were observed for the F_mean_non-dominant_ and dF.

Figure 2 shows a comparison of the propulsion time-series by SPM: Panel (A)—between junior and juvenile swimmers for the dominant upper limb; Panel (B)—between junior and juvenile swimmers for the non-dominant upper limb. SPM produced non-significant (*p* > 0.05) differences between all pairwise comparisons.

## 4. Discussion

This study aimed to compare swimming velocity and a set of kinematical variables between junior and juvenile swimmers. It also compared the propulsion outputs by discrete and continuous analyses (SPM) for each upper limb (i.e., dominant and non-dominant) between junior and juvenile swimmers. The main results indicated that junior swimmers presented a significantly faster swimming velocity than their juvenile counterparts. Based on the discrete analysis, higher F_mean_ was observed in both upper limbs in juniors compared to juvenile swimmers (not significant in the non-dominant). SPM (continuous analysis) showed non-significant differences between age groups for both upper limbs. Thus, the study hypotheses were partially incorrect.

The data indicate that the analyzed junior swimmers were faster than the juveniles. They also presented lower dv, faster SF, and, consequently, shorter SL. These findings agree with other studies, i.e., faster swimmers present lower dv [27] and faster SF [28]. Regarding SL, as swimming speed biomechanically depends on the interaction between SF and SL (v = SF ∙ SL), swimmers can exchange SF and SL during an event or trial. Indeed, whenever an increase in SF is observed, there is often a consequential tendency for SL to decrease [29]. However, it has recently been shown that this relationship can also be influenced by the generated propulsion [30]. The authors observed that the maximum swimming velocity was not reached at the highest SF or longest SL, nor with the largest propulsion [30]. In fact, this was achieved with an “optimal” SF–SL combination. Furthermore, it has been argued that increasing propulsion may not present a positive effect on swimming velocity if swimmers do not decrease drag [30].

As mentioned earlier, juniors presented faster swimming velocity than their juvenile counterparts. It was hypothesized that a key factor for this phenomenon would be the amount of propulsion generated, as greater propulsion leads to faster swimming velocities [8,14]. Specifically, it was speculated that junior swimmers would have greater and more significant propulsion than juveniles in both upper limbs. However, this assumption was not completely verified. Through discrete variables (*t*-test), the F_mean_dominant_ was higher and more significant in juniors than in juveniles. On the other hand, the F_mean_non-dominant_ and dF (both upper limbs) were higher, but not significantly. Based on the continuous analysis, SPM has proven to be a statistical approach that provides deeper insights into time-series data, providing point-to-point information [31,32]. For the stroke cycle, this can be of paramount importance based on the main key moments that characterize it [33]. Through continuous analysis, researchers and coaches can better understand in which specific moment of the stroke cycle a propulsion effect or difference can be verified. It was shown that propulsion changes in each key moment of the stroke cycle [34]. However, despite juniors having presented greater propulsion than juveniles in both upper limbs, SPM did not yield significant differences in the entire stroke cycle. The literature lacks evidence about these assumptions between age groups. Nonetheless, it was shown that in adult swimmers, the main trend is that greater propulsion leads to faster swimming velocities [35,36]. On the other hand, Koga et al. [8] noted that the fastest swimmer was not the one showing the most propulsion. Thus, despite being a key factor in increasing swimming velocity, greater propulsion alone cannot guarantee an increase in velocity.

Indeed, this topic was recently argued in age-group swimmers [30]. Swimmers must generate greater propulsion to increase swimming velocity, but they also must adopt the best possible streamline position during and after each arm pull, not only to generate propulsion but also to decrease drag [30]. It was noted that swimmers may suffer misalignments along their longitudinal axis when performing the arm pull, leading to a greater frontal surface area and, consequently, to greater drag [33,37]. As a result, despite generating greater propulsion, they may experience greater drag during or after the arm pull, which will decrease their velocity. Thus, the reason behind the fact that the fastest swimming velocity was achieved by junior swimmers may not be the propulsion alone (greater but not significant). It can be argued that drag, specifically active drag, could also be a determining factor for such a difference in swimming velocity. Despite having a larger frontal surface area, juniors can exhibit better swimming technique, allowing them to have better hydrodynamics. In age-group swimmers, it was observed that swimmers with larger frontal surface area were also the ones with the fastest swimming velocity, lowest active drag coefficient [38], and lowest intra-cyclic variation in swimming velocity [28]. These findings indicate that swimmers should not focus on generating greater propulsion if this achievement will lead to changes in the swimmer’s hydrodynamic position. Thus, coaches must design training programs for swimmers to increase propulsion and decrease drag. They should be aware that focusing on maximizing propulsion without considering swimmers’ hydrodynamics may not lead to an increase in swimming velocity.

Junior swimmers were significantly faster and presented greater values of propulsion than their juvenile counterparts. Usually, greater values of propulsion lead to faster swimming velocities. However, SPM (continuous analysis for time-series data) yielded non-significant differences in propulsion by the dominant and non-dominant upper limbs between junior and juvenile swimmers during the entire stroke cycle. This indicates that, at least for these age groups, the propulsion of both upper limbs was not the main differentiating factor for the swimmers’ competitive level. Thus, other factors, such as drag, may also be considered for this significant difference observed in swimming velocity. Coaches and swimmers should be advised that generating greater propulsion may not have the expected positive and direct effect on swimming velocity.

Overall, swimming researchers, coaches, and practitioners have been using video analysis as the mainstream procedure to assess swimming technique [4,39]. Currently, based mainly on technological development, the swimming community has alternatives that allow for the collection and analysis of data based on more accessible and user-friendly wearables [6]. In the case of propulsion, these wearables are based on pressure sensors [7] and inertial measurement units [40]. Notwithstanding, for studies that also aim to collect and analyze concurrently linear or angular kinematics, researchers and coaches are advised to use inertial measurement units as in terrestrial movements [41]. As the main limitations, it can be considered that: (i) these assumptions are for swimmers in these age groups and for maximum trials and (ii) the average of the three stroke cycles was used for analysis. In this sense, future research should focus on: (i) analyzing swimmers’ propulsion at different sub-maximum trials and with other age groups and (ii) analyzing the variance in propulsion (i.e., repeated measures) instead of the average, which may deliver meaningful insights to understand the swimmers’ intra-variability. Moreover, researchers and practitioners may also consider analyzing concurrently the propulsion and drag. As noted, swimming velocity was significantly different between groups, but propulsion was not. Therefore, it can be argued that whenever a significant difference is observed for swimming velocity but not for propulsion, the drag can be the reason behind the swimming velocity difference.

## 5. Conclusions

Junior swimmers showed faster swimming velocity than their juvenile counterparts, as well as faster SF and lower dv. Based on average values (discrete analysis), mixed results were found. A greater F_mean_ in both upper limbs was observed for juniors compared to juvenile swimmers, but only significant in the dominant one. On the other hand, the SPM (continuous analysis), as a statistical approach that provides deeper insights into times-series data, produced non-significant differences between age groups for both upper limbs. Thus, it can be stated that propulsion was not (at least not the only) key factor responsible for the significant difference in swimming velocity. Drag, during the arm pull or immediately after, can also play a determinant role on the swimming velocity difference.

## Figures and Tables

**Figure 1 sensors-22-08549-f001:**
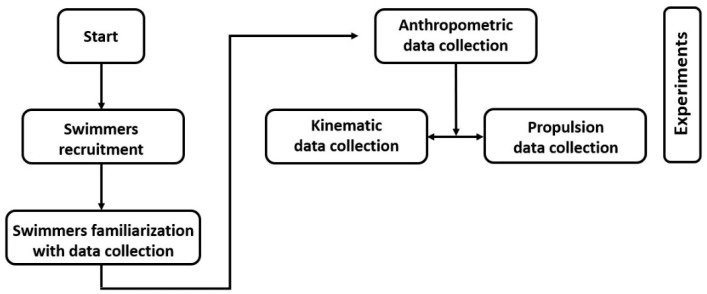
Flowchart of the experimental procedure.

**Figure 2 sensors-22-08549-f002:**
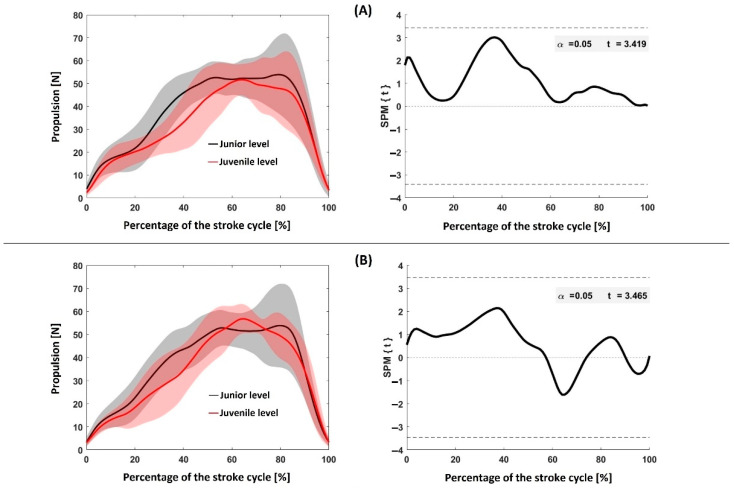
Comparison of propulsion–time series by SPM between and within groups. Panel (**A**): left side—mean and one standard deviation for the junior and juvenile dominant upper limb propulsion–time curve; right side—correspondent SPM comparison. Panel (**B**): left side—mean and one standard deviation for the junior and juvenile non-dominant upper limb propulsion–time curve; right side—correspondent SPM comparison. SPM {t}—*t*-test statistic for a statistical parametric mapping. Dashed lines represent the 95% confidence intervals (95CI).

**Table 1 sensors-22-08549-t001:** Descriptive data (mean ± one standard deviation) of all variables assessed for the two groups and the kinematics and propulsion comparison between groups.

	Junior Level (Mean ± 1 SD)	Juvenile Level (Mean ± 1 SD)	Mean Difference (95CI)	*t*-Test (*p*)	d [Descriptor]
**v [m∙s^−1^]**	1.66 ± 0.04	1.60 ± 0.09	−0.063 (−0.124 to −0.002)	−2.18 (0.04)	0.86 [moderate]
**dv [%]**	8.45 ± 1.85	11.56 ± 5.78	3.109 (−0.561 to 6.780)	1.77 (0.092)	0.72 [moderate]
**SF [Hz]**	0.89 ± 0.06	0.81 ± 0.08	−0.076 (−0.135 to −0.016)	−2.65 (0.015)	1.13 [moderate]
**SL [m]**	1.88 ± 0.14	1.98 ± 0.14	0.010 (−0.021 to 0.220)	1.72 (0.101)	0.71 [moderate]
**F_mean_dominant_ [N]**	37.14 ± 4.65	32.46 ± 5.39	−4.683 (−9.149 to −0.217)	−2.19 (0.041)	0.93 [moderate]
**dF_dominant_ [%]**	52.60 ± 10.53	50.36 ± 5.07	−2.245 (−9.852 to 5.363)	−0.62 (0.545)	0.27 [small]
**F_mean_non-dominant_ [N]**	36.27 ± 4.51	33.61 ± 5.55	−2.651 (−7.119 to 1.817)	−1.24 (0.230)	0.53 [small]
**dF_non-dominant_ [%]**	54.63 ± 9.51	51.95 ± 5.80	−2.683 (−9.874 to 4.508)	−0.78 (0.445)	0.34 [small]

v—swim velocity; dv—intra-cyclic variation of swim velocity; SF—stroke frequency; SL—stroke length; F_mean_dominant_—mean propulsion of the dominant upper-limb; dF_dominant_—intra-cyclic variation of propulsion of the dominant upper limb; F_mean_non-dominant_—mean propulsion of the non-dominant upper-limb; dF_non-dominant_—intra-cyclic variation of propulsion of the non-dominant upper limb; 95CI—95% confidence intervals; *p*—significance value; d—Cohen’s effect size.

## Data Availability

Not applicable.

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
