# Peer review of "Using Statistical Parametric Mapping to Compare the Propulsion of Age-Group Swimmers in Front Crawl Acquired with the Aquanex System"

_sensors, 2022, doi:10.3390/s22218549_

Round 1
Reviewer 1 Report
Dear Authors,
thank you very much for sending the article titled: "Using Statistical Parametric Mapping to compare age-group swimmers’ propulsion at front-crawl acquired with the Aquanex system". Generally, the paper is quite interesting to my mind, however, the authors should refer to the following statements:
- Please include a Flowchart of the experiment at the beginning Materials and Methods section
- The length of the article is shorter than the typical manuscripts at Sensors. I suggest it be titled as Letter and not an Article
- Please include in the article parameters of pressure sensors. It will be interesting to use in future Inertial Measurement Units in experiments. Look for example in the article titled: "A Kinematic Model of a Humanoid Lower Limb Exoskeleton with Pneumatic Actuators", Acta of Bioengineering and Biomechanics, where authors use this kind of device. You can obtain kinematic parameters (e.g. angles, angular velocities and acceleration part of the body and compare with pressure sensors). It would be interesting. To improve article quality, please cite above manuscript and write a few sentences in Discussion.
Author Response
REV #1
Dear Authors,
thank you very much for sending the article titled: "Using Statistical Parametric Mapping to compare age-group swimmers’ propulsion at front-crawl acquired with the Aquanex system". Generally, the paper is quite interesting to my mind, however, the authors should refer to the following statements.
Authors: Thank you very much for the time you spent and your constructive feedback on this manuscript. We have made every effort to take on board your recommendations and comments. We hope this 2nd revised version and the responses to the comments (kindly refer to our replies below) will meet your requirements. Please note that all new changes in the revised manuscript are highlighted in yellow.
- Please include a Flowchart of the experiment at the beginning Materials and Methods section
Authors: This was added as suggested.
- The length of the article is shorter than the typical manuscripts at Sensors. I suggest it be titled as Letter and not an Article
Authors: We changed this as advised. However, we think that the editor will have a final word about this.
- Please include in the article parameters of pressure sensors. It will be interesting to use in future Inertial Measurement Units in experiments. Look for example in the article titled: "A Kinematic Model of a Humanoid Lower Limb Exoskeleton with Pneumatic Actuators", Acta of Bioengineering and Biomechanics, where authors use this kind of device. You can obtain kinematic parameters (e.g. angles, angular velocities and acceleration part of the body and compare with pressure sensors). It would be interesting. To improve article quality, please cite above manuscript and write a few sentences in Discussion.
Authors: We appreciate the reviewer’s comment. This study was added in the reference’s section. Also, we added some information at the end of the discussion section about this topic as advised.
Reviewer 2 Report
Below is my feedback
Introduction:
The introduction is well-written except there are some very minor errors (i.e. "On the other hand, less evidence can be found about the swimmers' propulsion directly measured")
Methods:
For readers who are unaware of the differences between junior and juvenile level, can you please identify the differences between the two levels.
How were swimmers recruited?
What were the inclusion/exclusion criteria?
Was there an a priori power analysis conducted prior to the study?
How many swimmers were recruited?
How many total trials were conducted? You state that you recorded the fastest trial, but didn't talk about how many trials were conducted
How much time elapsed between trials?
There were also minor grammatical errors.
Did you assess for demographic differences between the groups?
Results
There were several minor grammatical errors in this section as well
You stated that you examined dominant versus non-dominant hand however, you did not identify that in your methods. How did you identify dominant hand? Was it self-report?
Why not just put the results of Table 2 in Table 1 and just have 1 table?
While I appreciate the Figures there were color issues with identifying junior level versus juvenile. Since this is an Open Access online journal, I would suggest using color. It would help the figures significantly.
Discussion
Once again there are several grammatical errors here
You bring up some hypotheses that were never explained in the introduction. Please provide those in the introduction
What were the limitations of your study?
What are the future directions for this work?
Author Response
REV #2
Thank you very much for the time you spent and your constructive feedback on this manuscript. We have made every effort to take on board your recommendations and comments. We hope this 2nd revised version and the responses to the comments (kindly refer to our replies below) will meet your requirements. Please note that all new changes in the revised manuscript are highlighted in yellow. Please note that a native English speaker reviewed this manuscript’s final version as advised.
Introduction:
The introduction is well-written except there are some very minor errors (i.e. "On the other hand, less evidence can be found about the swimmers' propulsion directly measured")
Authors: We appreciate the reviewer’s comment. This was edited as advised. As aforementioned, a native English speaker reviewed this manuscript’s final version as advised.
Methods:
For readers who are unaware of the differences between junior and juvenile level, can you please identify the differences between the two levels.
Authors: We appreciate the reviewer’s comment. The levels are based on their age-group competition (i.e., chronological age). This was added as advised for clarity’s sake.
How were swimmers recruited?
Authors: Swimmers were recruited from the national squad team of their respective competitive level age-group during a swim camp. This was added as advised.
What were the inclusion/exclusion criteria?
Authors: This was added as advised.
Was there an a priori power analysis conducted prior to the study?
Authors: We did not perform such analysis. However, please note that we are dealing with the best male sprint performers in their nation age-group. We must feel flattered by having the chance to analyze these swimmers, specially because coaches not always agree with this extensive analysis.
How many swimmers were recruited?
Authors: We recruited the male swimmers specialized in sprinting freestyle events (50 and 100m) that were participating in the swim camp (juniors: N = 12; juveniles: N = 10). This is reported in participants’ section.
How many total trials were conducted? You state that you recorded the fastest trial, but didn't talk about how many trials were conducted
Authors: We apologize for this. Three trials were conducted. This was added as advised.
How much time elapsed between trials?
Authors: A 10-minute interval between trials was given to recovery. This was added as advised.
There were also minor grammatical errors.
Authors: We apologize for this. A native English speaker reviewed the final version of the manuscript.
Did you assess for demographic differences between the groups?
Authors: The demographics assessed between groups are mentioned in the participants’ section: age, body mass, height, arm span, frontal surface area and FINA points. As non-significant differences were observed in the swimmers’ demographics, we did not report this in table 1. Significant differences were only noted in the swimmers’ age, which is the factor for the competitive level pooling. This was added as previously mentioned.
Results
There were several minor grammatical errors in this section as well
Authors: We apologize for this. A native English speaker reviewed the final version of the manuscript.
You stated that you examined dominant versus non-dominant hand however, you did not identify that in your methods. How did you identify dominant hand? Was it self-report?
Authors: We appreciate the reviewer’s comment. Indeed, this was performed by self-report. We added this as advised.
Why not just put the results of Table 2 in Table 1 and just have 1 table?
Authors: This was done as advised.
While I appreciate the Figures there were color issues with identifying junior level versus juvenile. Since this is an Open Access online journal, I would suggest using color. It would help the figures significantly.
Authors: We agree with the reviewer’s comment. The figures were changed for a color setting as advised.
Discussion
Once again there are several grammatical errors here
Authors: We apologize for this. A native English speaker reviewed the final version of the manuscript.
You bring up some hypotheses that were never explained in the introduction. Please provide those in the introduction
Authors: We understand the reviewer’s point of view. However, please let us explain ours. The focus of this manuscript is the propulsion generated by the swimmers and the statistical approach used to compare both groups. In the intro section we hypothesized that propulsion would have a significant contribution into the groups’ difference (since it is one of the two main factors). And based on the stats used, it would be possible to identify where within the stroke cycle. However, our results indicated that significant differences were observed in velocity, but not in propulsion (at least significantly, since juniors generated greater levels of propulsion than juveniles). Thus, another factor (alone or combined with propulsion) may be responsible for the swim velocity difference (i.e., drag). But we only acknowledge that after the results. As the reviewer can see, we did mention that swim velocity depends on the net balance between drag and propulsion. Here, we already mention the two main factors responsible for the swim velocity enhancement. However, as aforementioned, our mainframe was about propulsion. We now added minor information about drag as advised in the intro section, but this drag topic can only be highlighted after the results. Thus, we hope the reviewer agrees with us, and this drag hypothesis only comes-up in the discussion section. Specially, because it is related to one of your next comments: the future directions.
What were the limitations of your study?
Authors: Please note that limitations are already given at the end of the discussion section: “As main limitations it can be considered that: (i) these assumptions are for these age-group swimmers and for maximal trials, and; (ii) the average of the three stroke cycles was used for analysis”.
What are the future directions for this work?
Authors: We appreciate the reviewer’s comment. We already have a sentence about this, but we elaborated a bit more as advised.
Round 2
Reviewer 2 Report
I appreciate the authors addressing all of my concerns.